# Modularity is the Bedrock of Natural and Artificial Intelligence

## Abstract

The astonishing performance showcased by AI systems in the last decade has been achieved through the use of massive amounts of data, computation, and, in turn, energy, which vastly exceed what human intelligence requires. This wide gap underscores the need for further research and points to leveraging brains as a valuable source of guiding principles. On the other hand, the No Free Lunch Theorem highlights that effective inductive biases must be problem-specific. This suggests designing architectures with specialized components that can solve subproblems — namely, modular architectures. Interestingly, modularity is an established principle of brain organization that is considered essential for supporting the efficient learning and strong generalization abilities consistently demonstrated by humans. However, despite its recognized importance in natural intelligence and the proven benefits it has shown across various seemingly unrelated AI research areas, modularity remains somewhat underappreciated in AI. In this work, we review several research threads in artificial intelligence and neuroscience through a conceptual framework that highlights the central role of modularity in supporting both artificial and natural intelligence. In particular, we examine what computational advantages modularity provides, how it has emerged as a solution in several AI research areas, which modularity principles the brain exploits, and how modularity can help bridge the gap between natural and artificial intelligence.

## Table of Contents

# 1 Introduction

## 1.1 Capabilities and Limitations of Current AI Systems

Present-day AI systems can execute well-defined tasks with a proficiency that appeared unattainable just a few years back. Convolutional Neural Networks (CNNs) (He et al., 2016) classify images with superhuman accuracy (Russakovsky et al., 2015); Reinforcement Learning (RL) agents can defeat elite professional players in complex strategy games with intractable search spaces, such as Go (Silver et al., 2016); Large Language Models (LLMs) excel at natural language processing tasks (Brown et al., 2020), master professional and academic knowledge (Achiam et al., 2023), coding[1][2], and appear[3] to be able to perform abstract reasoning (Chollet, 2019).

However, these feats are accomplished with massive quantities of data, computation, and, in turn, energy, which vastly exceed what human intelligence necessitates. For example, CNNs can reach high classification accuracy only after being trained on thousands of examples for each class (LeCun et al., 1998; Russakovsky et al., 2015), whereas humans can do so even after being exposed to a single example (Lake et al., 2015). To make things worse, CNNs typically have to be presented with several variations of the same instance over different epochs through data augmentation. Similarly, RL agents, such as AlphaGo (Silver et al., 2016), can achieve superhuman performance only after being trained for at least three orders of magnitude more games than elite human players, who excel in a much wider array of tasks (Lake et al., 2017). Finally, the impressive skills of current LLMs require datasets of tens of trillions of tokens (Dubey et al., 2024), which would take an above-average reader (Brysbaert, 2019) 5-50 thousand years of continuous reading. This poor sample efficiency is accompanied by poor energy efficiency: training a model such as GPT-3 (Brown et al., 2020) was estimated to consume 1287 megawatt hours (MWh), an amount of energy that would power over 100 average American households for a year — roughly three orders of magnitude more than the 3.15 MWh required to power a 20W human brain (Sokoloff, 1960) for 18 years. The energy consumption of next-generation LLMs, and thus their resulting carbon footprint, is likely to worsen further over the coming years as more recent models are rumored to have at least 10X more parameters[4] and rely heavily on additional test-time computations (Yao et al., 2024; Snell et al., 2024; Guan et al., 2025) due to the emerging performance gains[3].

## 1.2 The Relevance of Modularity Principles as a Path Forward

Despite their reliance on enormous datasets and energy resources, current AI systems typically still struggle with out-of-distribution (OOD) generalization (Geirhos et al., 2020; Yuan et al., 2023; Mayilvahanan et al., 2024), have poor compositional skills (Lake & Baroni, 2018; Hupkes et al., 2020; Dziri et al., 2024), and suffer from catastrophic forgetting and limited positive transfer (French, 1999; Luo et al., 2023; Huang et al., 2024), while humans, typically, do not (Lake et al., 2017; Lake & Baroni, 2018; Ito et al., 2022). How can we move beyond such limitations? The No Free Lunch Theorem (Wolpert et al., 1995; Wolpert & Macready, 1997) suggests that no inductive bias can lead to the design of a general-purpose architecture that is a universal problem solver. Thus, good inductive biases need to be problem-specific (Sinz et al., 2019). It follows that rather than attempting to build monolithic architectures with good inductive biases, we should focus on building problem-specific modules. Therefore, tackling some of the limitations of current AI systems by leveraging *modularity principles* — whereby specialized modules work synergistically to solve complex tasks requiring new combinations of learned skills — appears to be a particularly promising research avenue.

Modular architectures were indeed proposed early on (Grossberg, 1976; Rueckl et al., 1989; Jacobs et al., 1991a;b; Happel & Murre, 1994) as a way to both capture the modular design of brains and speed up learning, and they were shown to perform competently while reproducing selected features of brain activation patterns. Currently, however, the usage of modular designs is driven by two main motivations: the growing need to efficiently reuse large pre-trained models across different domains while minimizing the costs associated

---

[1] https://www.anthropic.com/news/claude-3-5-sonnet
[2] https://openai.com/index/learning-to-reason-with-llms/
[3] https://arcprize.org/blog/oai-o3-pub-breakthrough
[4] https://en.wikipedia.org/wiki/GPT-4

with fine-tuning, and the ambition to design architectures capable of performing multi-task and, ideally, continual learning (Pfeiffer et al., 2023; Yadav et al., 2024). Nevertheless, a closer look reveals that modular architectures have also emerged as a powerful solution to diverse challenges across various AI research areas, suggesting a broader convergence toward modular designs (Andreas et al., 2016; Rusu et al., 2016; Guo et al., 2025). Moreover, modularity principles are virtually omnipresent in modern AI systems, as deep neural networks (DNNs) have an inherently modular structure composed of stacked layers — an inductive bias that grants them significant computational advantages (Poggio et al., 2017). Interestingly, as we illustrate in § 4, brains are also modular (Fodor, 1983; Simon, 1962; Meunier et al., 2009), and modularity is believed to be core to the learning efficiency and robust generalization abilities humans possess (Zhang et al., 2024).

### 1.3 Scope and Outline

In this work, we survey and synthesize a large body of research from fields spanning artificial intelligence, neuroscience, evolutionary biology, complex systems, and engineering theory, highlighting how modularity consistently emerges as a key computational principle across all these scientific domains. We demonstrate how modularity has been primarily studied in isolation in each of these fields, resulting in associations with distinct properties and functions. For instance, in complex systems, modularity has been linked to robustness and network communication efficiency, while in evolutionary biology, it has been linked to evolvability and adaptability. In neuroscience, it is often associated with energy efficiency and functional specialization at multiple organizational levels, from neurons to brain networks. Finally, in artificial intelligence, modularity is often studied as an implicit structural property (e.g., layers in deep neural networks), an explicit architectural feature (e.g., mixture of experts in large language models), and also as an emergent property in highly performing networks (e.g., induction heads in large language models). These observations lead us to argue that modularity is not just a recurring feature in different scientific domains but constitutes a fundamental computational principle common to both natural and artificial intelligence. Thus, it might serve as a core design principle in future AI systems and a valuable focus for future research. Furthermore, because modular architectures can be challenging to design effectively, we propose that insights from the brain may help identify the fundamental functions that modules should specialize in, offering a promising avenue for bridging the gap between natural and artificial intelligence.

In the remainder of this work, we first discuss why modularity is a fundamental design principle in engineering (§ 2.1) with clear robustness advantages. We then present evidence of its widespread presence in complex natural systems (§ 2.2). Next, we introduce a formalism to describe modular AI frameworks (§ 2.3), review influential work that sheds light on the computational advantages they provide (§ 3.1,§ 3.2), and examine their growing adoption across several AI research areas (§ 3.3). We then survey key modularity principles the brain is thought to exploit (§ 4), at spatial scales ranging from neurons (§ 4.1) and circuits (§ 4.2) to cortical networks (§ 4.3). Finally, we discuss open research questions (§ 5) — touching on topics such as the continued relevance of brains for the development of AI systems (§ 5.2) and the role of modularity to help bridge the gap (§ 5.3) — and summarize the main takeaways of this work (§ 6).

## 2 Modularity Principles

A system is modular if it is composed of subsystems whose structural elements are strongly connected among themselves and weakly connected to elements of other subsystems (Rumelhart et al., 1986; Baldwin & Clark, 1999). In a typical modular system, the modules are *specialized* — i.e., excel at performing specific functions — *sparsely interacting* — i.e., can exchange information when needed — and *largely autonomous* — i.e., do not rely on other modules to perform their functions. This organization naturally implements a divide-and-conquer strategy, whereby a complex problem is decomposed into sub-problems that modules are specialized in solving. This decomposition is achieved through *information factorization*, which fosters both specialization and robustness: since the modules receive only information that is relevant to perform their function, they are invariant to (and thus robust against) irrelevant information and can effectively specialize in processing their inputs (Merel et al., 2019).

| Evolutionary Biology | Neuroscience | Artificial Intelligence | Engineering Theory |
|---|---|---|---|
| Robustness | Energy efficiency | Compositional generalization | Splitting |
| Adaptability | Functional specialization | Continual learning | Substituting |
| Evolvability | Information factorization | Transfer learning | Augmenting |
| | Partial autonomy | Multi-task learning | Excluding |
| | Dynamical richness | Amortized learning | Inverting |
| | Timescale separation | Multi-modal learning | Porting |
| | | Neuro-symbolic learning | |

Table 1: Core properties and capabilities that modular architectures are theorized to promote in different scientific domains.

## 2.1 Modularity in Engineering

Modularity principles are widely applied in engineering (Suh, 1990), particularly in software (Booch et al., 2008) and hardware (Baldwin & Clark, 1999) design, and are regarded as foundational to building scalable and robust systems (Lipson et al., 2007). A good overview of the properties underlying their success in engineering is provided by the six fundamental *operators* identified by Baldwin & Clark (1999): splitting, substituting, augmenting, excluding, inverting, and porting. *Splitting* allows breaking a complex problem into simpler, module-specific, sub-problems. *Substituting* enables replacing one module with an upgraded version. *Augmenting* allows adding new modules to the system (providing either new functionalities or enhanced robustness through the introduction of redundancy). *Excluding* allows removing unnecessary modules. *Inverting* facilitates the creation of better design rules that leverage the existing modules more efficiently. *Porting* allows reusing existing modules in new systems. These unique properties of modular systems are often considered fundamental to the evolution of technology, fostering a continuous cycle of refinement where existing components are incrementally improved or new ones are seamlessly integrated into the systems.

## 2.2 Modularity in Nature

### 2.2.1 Ubiquity and Robustness

Modularity principles, as we have argued in the previous section, are widespread in artificial systems as they foster the design of scalable, robust, and increasingly sophisticated systems. Is there evidence of similar principles also in the natural world? It turns out that most complex systems[5], such as biological, physical, social, and symbolic systems are also widely regarded as being modular, with modules often arranged in hierarchies (Simon, 1962; Callebaut & Rasskin-Gutman, 2005; Newman, 2006). For example, vertebrates can be understood as organ systems, where each system performs a specialized function. These systems are composed of organs, which carry out sub-functions. Moving further down the hierarchy, we encounter tissues, cells, organelles, proteins, polypeptides, amino acids, etc. At every level of this hierarchy, we observe systems composed of modules, each often dedicated to specialized functions.

It thus appears that modularity is a widespread organizational principle in natural systems. This raises the question: what underlies its ubiquity? It has been theorized that the reason why most complex systems are hierarchically modular is because this organization promotes the formation of relatively stable, long-lived, intermediate units — the modules — that mitigate the natural tendency toward disorder acting on their constituent parts (Schrödinger & Penrose, 1992; Simon, 1962). As these intermediate units typically exhibit emergent functionalities and will also tend to aggregate into stable super-units, systems will tend toward greater complexity and sophistication. Consistent with this, graph-theoretical analyses of wide ranges of complex natural networks—including brain networks and social interaction networks—often reveal strong signatures of small-world and modular organization. These networks achieve strong local specialization and efficient global communication through their dense local three-node interactions, efficient long-range shortcuts that connect distant nodes in the network, and communities of specialized modules that interact smoothly, leveraging tightly linked connector hubs (Watts & Strogatz, 1998; Sporns & Betzel, 2016).

---

[5]Simon (1962) considered complex all the systems composed of multiple interacting parts interacting in non-trivial ways

### 2.2.2 An Evolutionary Perspective: Evolvability and Adaptability

The modularity of biological systems (Wagner et al., 2001; 2007) has been typically studied also in relation to evolution and has been hypothesized to favor *evolvability*: the ability to generate and propagate advantageous phenotypic traits (Kirschner & Gerhart, 1998). According to this theory, modular designs allow selective pressure to optimize each module separately without interference (Hansen, 2003). In fact, this design renders disadvantageous mutations affecting individual modules less likely to be lethal, and advantageous mutations more likely not to be associated with negative pleiotropic effects. Influential simulation studies identified additional mechanisms underlying the origin of modularity in biological systems. These studies suggested that modularity in biological networks arises in response to changing environments (Lipson et al., 2002), that this effect is particularly strong when the environment changes in a modular manner (i.e., keeping sub-requirements constant Kashtan & Alon (2005)), and that it significantly speeds up *adaptation* to new environments (Kashtan et al., 2007). Interestingly, this finding may explain why more recent studies have found that multi-task learning promotes modularity in CNNs (Dobs et al., 2022) and RNNs (Yang et al., 2019). Additional studies highlighted that modularly changing environments are not the only potential determinant of modular networks, as the minimization of connection costs in constant environments also leads to modular solutions (Clune et al., 2013).

### 2.2.3 Energetic and Computational Benefits of Modular Brain Networks

Finally, we note that while modular networks have been studied in the context of evolution, they have also served as powerful models for understanding the computational principles of natural intelligence. For instance, studies suggest that optimizing artificial neural networks with a bias toward short connections — which offer clear energetic benefits due to the cost of transmitting information — naturally leads to modular networks. These networks decompose tasks into subtasks (Jacobs & Jordan, 1992), exhibit greater resilience to catastrophic forgetting in continual learning (Ellefsen et al., 2015), demonstrate brain-like mixed selectivity and low average activation patterns (Achterberg et al., 2023), and form sparse information streams while reusing useful features (Liu et al., 2023). Taken together, these findings highlight some of the computational advantages that brains may rely on due to their modular organization (Meunier et al., 2009).

Other key computational benefits of modular brain networks are discussed in (Merel et al., 2019). Modules typically act on specific projections of their inputs, making them robust to variations orthogonal to those projections and enabling specialization in processing the task-relevant projections they receive—a property referred to as *information factorization*. Sparse inter-module connectivity further supports *partial autonomy*, allowing modules to function without constant coordination or supervision from other modules, thereby containing the impact of local failures. Partial autonomy also facilitates *amortized control*: modules with complementary roles can be specialized such that fast, low-cost reactive modules handle most situations, while more computationally expensive learning modules are recruited only when reactive strategies fail. Once a new solution is acquired, it can be transferred back to reactive modules for future use (c.f. § 3.3.4). Finally, module specialization enables *timescale separation*, which allows networks to operate across multiple timescales, with some modules setting long-horizon goals while others manage short-term actions. In general, timescale separation (Arenas et al., 2006; Pan & Sinha, 2009), together with phenomena such as multistability and metastability, are key features of the *complex dynamics* that modular architectures have been shown to promote (Sporns et al., 2000; Palma-Espinosa et al., 2025). Thus, modular networks appear to provide the right substrate for generating the complex high-dimensional dynamics that are the core component of reservoir computing (Jaeger, 2001; Maass et al., 2002; Sumi et al., 2023).

### 2.3 A Formal Definition

A more formal definition of a modular model in machine learning is as follows. Given an input $x$, a model $f(x) : \mathbb{R}^i \to \mathbb{R}^o$ is modular with modules $\mathcal{M} = \{m_{\mu_i}(x)\}_{i=1}^M$ if it can be rewritten as:

$$f(x) = \phi(m_{\mu_1}(x), m_{\mu_2}(x), ..., m_{\mu_M}(x)) \tag{1}$$

where $m_{\mu_i}(x)$ is the $i$-th module with parameters $\mu_i$. Typically, modular models have a *routing function* $r_\rho(x) : \mathbb{R}^i \to 2^{\mathcal{M}}$ with parameters $\rho$, which determines which modules are active, and an *aggregation function*

| Implicit | Emergent | Architectural |
|---|---|---|
| Units in NNs | Functional modules in multi-task learning | Experts in MoEs |
| Layers in DNNs | Knowledge neurons in LLMs | RAG DBs in industrial chatbots |
| Winning tickets in LTH | Induction head circuits in LLMs | Agents in multi-agent systems |

Table 2: A proposed taxonomy of modularity in artificial intelligence with representative examples. Implicit modules are intrinsic to virtually all modern architectures, emergent modules arise naturally during training, and architectural modules are explicitly designed for specialization.
Abbreviations: NNs = Neural Networks; DNNs = Deep Neural Networks; LTH = Lottery Ticket Hypothesis; LLMs = Large Language Models; MoEs = Mixture of Experts; RAG DBs = Retrieval-Augmented Generation Databases.

$g_\gamma(x) : 2^{\mathcal{M}} \to \mathbb{R}^o$ with parameters $\gamma$, which determines how the modules are aggregated. Thus, we can rewrite a modular model more compactly as:

$$f(x) = g_\gamma(r_\rho(x)) \tag{2}$$

This notation also accommodates more sophisticated inter-module interactions, including hierarchical compositions $y = f_{l+1}(f_l(x))$ and recurrent formulations $x_{t+1} = f(x_t)$. Typically, in modular networks, different tasks elicit distinct network behavior, so the input $x$ often includes task information (e.g., it might be a concatenation of input features and task embeddings). Also, note that modules may operate on different input projections. Here, we assume these projections are extracted within the modules; however, they might also be extracted by the routing function.

Routing can be *hard* — when the modules are either active or inactive — or *soft* — when all modules are active with probability $p_i$. Critically, hard routing leads to sparse models, which are particularly efficient during inference as signals only need to propagate through selected modules. However, these models cannot be trained end-to-end via gradient descent and require specialized training techniques such as reinforcement learning (e.g., Rosenbaum et al. (2017)), evolutionary algorithms (e.g., Fernando et al. (2017)), or stochastic parametrization (e.g., Sun et al. (2020)). On the other hand, models with soft routing can be trained end-to-end via gradient descent but are not sparse and thus can be fairly computationally expensive.

Aggregation functions often define simple operations, such as the weighted summation: $g_\gamma(x) = \sum_{j \in r_\rho(x)} \alpha_j \, m_{\mu_j}(x)$ (as in Jacobs et al. (1991b); Shazeer et al. (2017)); however, in some other cases, they can also define more complex, attention-based operations, for example, with the input $x$ as a query, and the active modules' outputs $Z$ as key and values: $g_\gamma(x) = \text{Attn}(xQ, ZK, ZV)$, where $Q$, $K$, and $V$ are matrices of learned parameters (e.g., as in Pfeiffer et al. (2020a)).

Finally, we note that, in some contexts, such as efficient transfer learning (Pfeiffer et al. (2023), cf. § 3.3.3), modules are employed to fine-tune pre-trained models; in this scenario, they are interspersed between the layers of the base model to modify their behavior. In this case, one needs to also define a *modifier* function $d_\delta^{(f)}(\cdot)$ to specify how the modular model $f$ modifies the behavior of the base network's layer $l_\lambda(x)$. In this case, modifier functions tend define simple operations such as summations: $d_\delta^{(f)}(l_\lambda(x)) = l_\lambda(x) + f(x)$ (as in Ansell et al. (2021)), and function compositions $d_\delta^{(f)}(l_\lambda(x)) = f(l_\lambda(x))$ (as in Rebuffi et al. (2017)) in order to minimally alter the base network's behavior.

We refer the reader to Pfeiffer et al. (2023) for a more in-depth overview of the routing, aggregation, and modifier functions used in modular models.

# 3 Modularity in Artificial Intelligence

In the previous sections, we have presented evidence that modularity is a fundamental feature of complex natural systems, conferring numerous advantages. We have also shown how this insight has inspired engineers to formalize the benefits of modularity principles and harness them to design more robust artificial systems. Here, we demonstrate that these principles have also had a strong influence on AI.

In fact, modularity was an essential design feature of early-stage connectionist (Jacobs et al., 1991a), cybernetic (Wiener, 2019) and symbolic (Kautz, 2022) AI systems, which has more recently reemerged as a

central theme in several AI research areas such as Continual Learning, Transfer Learning, LLMs, RL, and Autonomous Agents. This trend is motivated by the recognition that modularity principles offer a promising means of increasing the efficiency and capability of current deep-learning-based AI systems while minimizing inter-task interference. Thus, modularity principles have been adopted, advocated, and reviewed in several influential publications, which we will discuss in this section.

We structure this section around a proposed **taxonomy of modularity** in artificial intelligence, comprising three categories: implicit, emergent, and architectural modularity (Table 2). **Implicit modularity** refers to the form of modularity that is intrinsic to Deep Neural Networks (DNNs) and is thus present in virtually all modern architectures, due to their organization into units and layers which compute specialized functions of their inputs. **Emergent modularity** refers to the form of modularity that arises naturally during training, for example, when units organize into structural or functional subcomponents. **Architectural modularity** characterizes architectures explicitly designed with modularity priors, encouraging the use of separate computational building blocks with specialized functions.

## 3.1 Implicit Modularity

Deep Neural Networks (DNNs) are composed of a stack of non-linear layers, where each layer provides a processed version of its input to its downstream layer. Thus, in essence, DNNs have a particular kind of modular architecture, a hierarchical architecture with layers — that is, its modules — arranged hierarchically. Thus, this design encourages information factorization (Merel et al., 2019): each layer tends to only receive the information that is relevant to compute its output and can specialize in learning this mapping. Empirically, it has been observed that hierarchical designs facilitate the extraction of progressively more complex, invariant, and abstract features along the hierarchy. For example, moving down the layers of a CNN, (Zeiler & Fergus, 2014; Olah et al., 2017), it is typical to find layers whose neurons are strongly activated by increasingly more global and abstract image features, going from edges, simple shapes, and textures to patterns, object parts, and entire objects[6].

Although DNNs took time to gain traction due to the lack of efficient training algorithms, large datasets, and efficient hardware (Goodfellow, 2016), the computational advantages of hierarchical architectures have long been known (Håstad, 1986). Influential work clarified that, although shallow architectures such as 1-hidden layer neural networks and kernel machines are also universal function approximators, they are inefficient learners of *highly varying functions* compared to DNNs due to their reliance on local estimators (Bengio & LeCun, 2007). More recent work (Poggio et al., 2017) has narrowed down the class of functions that DNNs excel at capturing to those with a compositional structure — that is, functions that can be written as a function of functions:

$$f(x) = h_L(h_{L-1}(...h_1(x)))$$ (3)

Specifically, DNNs are proven to avoid the curse of dimensionality when the function they are trained to approximate is compositional. Conversely, shallow neural networks can achieve the same approximation error only with a number of parameters that grows exponentially with the number of inputs. For example, compositional functions of $n = 8$ variables and smoothness $m$ with a binary tree-like computational graph, defined by

$$f(x_1, ..., x_8) = h_{1:8}(h_{1:4}(x_1, ...,x_4), h_{5:8}(x_5, ...,x_8))$$

$$h_{1:4}(x_1, ...,x_4) = \phi_{1:4}(h_{1:2}(x_1,x_2), h_{3:4}(x_3,x_4))$$ (4)

$$h_{5:8}(x_5, ...,x_8) = \phi_{5:8}(h_{5:6}(x_5,x_6), h_{7:8}(x_7,x_8))$$

can be approximated with an accuracy of at least $\epsilon$ by a shallow neural network with $N_{shallow} = \mathcal{O}(\epsilon^{-n/m})$ units, or by a deep network with $N_{deep} = \mathcal{O}((n-1)\epsilon^{-2/m})$ units[7].

While layers are the most widely recognized modules of DNNs, they are not the only ones. Layers themselves are composed of units, which can also be regarded as implicit modules when examined closely. Each unit

---

[6]Note that hierarchical processing is gradual, with adjacent layers performing similar functions especially in very deep networks (Lad et al., 2024; González et al., 2025)

[7]This statement is true as long as the graph defining $f(\cdot)$ is a subgraph of the graph defining the DNN

computes a specific function of its inputs, producing transformations that help extract informative features from the data. For instance, units in convolutional layers act as feature detectors, responding to the presence of particular patterns within their receptive fields.

More recently identified DNN components that can also be regarded as implicit modules are the *winning tickets* described by the *Lottery Ticket Hypothesis* (LTH) (Frankle & Carbin, 2018). According to the LTH, dense, randomly initialized DNNs contain sparse subnetworks that can be trained to achieve performance comparable to the original dense networks. These subnetworks—the winning tickets—are specialized components of the dense model and can thus be regarded as modules. Subsequent studies showed that winning tickets can generalize across datasets (Morcos et al., 2019), and that sufficiently overparameterized random DNNs contain multiple such tickets capable of solving a task with comparable accuracy to the full dense network, even without additional training (Diffenderfer & Kailkhura, 2021). Taken together, these findings suggest that such subnetworks are critical components of DNNs and that learning in DNNs may amount to identifying a suitable subnetwork, or module, for the task at hand.

### 3.2 Emergent Modularity

Recent studies have investigated emergent modularity, that is, the emergence of modular structures in DNNs that were not imposed by design. This property is of particular interest as the presence of modules helps in understanding the computational mechanism the networks use to solve the task. Ideally, by studying the activation patterns of the modules, one can identify the fundamental subfunctions or rules that are learned and exploited by the networks to perform the task. Importantly, when the networks learn compositionally — that is, when they learn the fundamental atomic rules underlying the target tasks and can combine them in arbitrary new ways (Fodor & Pylyshyn, 1988; Hupkes et al., 2020) — these subfunctions are systematically reused whenever the corresponding subtasks need to be performed. For example, a network that learns to classify objects compositionally can correctly label an image of a white cube even if, during training, it has only seen white spheres and red cubes, and, moreover, it does so using separate color and shape modules.

Traditionally, modules are identified by clustering connectivity (Watanabe et al., 2018; Casper et al., 2022) or activation (Watanabe, 2019; Lange et al., 2022) statistics, which makes understanding the functional role of a module very hard, and studying the compositionality of the network even harder. More recent studies (Csordás et al., 2020; Lepori et al., 2023a) were able to identify modules responsible for specific subtasks by training binary masks on the full network. Specifically, this approach brought to light subfunction-specific subnetworks. However, further analyses (Csordás et al., 2020) showed that the identified subnetworks were not reused in different contexts where the same rules had to be applied in a different combination, providing evidence of the lack of compositionality (Lake & Baroni, 2018; Barrett et al., 2018; Hupkes et al., 2020). Thus, DNNs might generally struggle to identify similarities between subtasks, an issue related to the more general problem of information binding (Greff et al., 2020).

A closely related line of work is the emerging *mechanistic interpretability* research field, which often focuses on identifying specialized circuit components that are reused across tasks with similar structures. Such studies typically analyze pretrained Large Language Models (LLMs) (Dai et al., 2021; Wang et al., 2022; Olsson et al., 2022; Conmy et al., 2023; Merullo et al., 2023; Lepori et al., 2023b), and have shown that these specialized components can endow models with a degree of compositional ability, especially at larger scales (Xu et al., 2024). Notable examples of emergent modules identified in this line of research include *knowledge neurons* (Dai et al., 2021), which activate strongly and selectively when specific pieces of information are recalled, and *induction head circuits* (Olsson et al., 2022), which mediate in-context learning.

### 3.3 Architectural Modularity

#### 3.3.1 Compositional Generalization

Several studies have shown that modular architectures possess better generalization ability (Clune et al., 2013; Andreas et al., 2016; Kirsch et al., 2018; Chang et al., 2018; Goyal et al., 2019; Mittal et al., 2022) and sample efficiency (Bahdanau et al., 2018; Purushwalkam et al., 2019; Khona et al., 2023; Boopathy et al., 2025) than their monolithic counterparts. A potential explanation for these advantages is their superior

compositional learning ability — i.e., their ability to learn and combine atomic rules in arbitrary new ways (Fodor & Pylyshyn, 1988; Hupkes et al., 2020). This skill, which humans excel at, is a significant challenge for AI systems (Lake & Baroni, 2018; Keysers et al., 2019) that modularity principles can help tackle (c.f., § 3.2). This has led to research work aimed at understanding how to best design modular architectures that can generalize compositionally.

In-depth analyses have demonstrated that modular networks can indeed achieve strong compositional generalization abilities (Andreas et al., 2016; Bahdanau et al., 2018); however, this only happens when the task structure is known and can be used to assign modules to the constituent subtasks they can specialize in (Bahdanau et al., 2018; Béna & Goodman, 2021; Mittal et al., 2022). However, it has been observed that when simulating real-world settings more closely, where the task structure is unknown, modular networks often do not specialize and do not exhibit a consistent performance boost. Thus, to fully leverage the capabilities of modular architectures, it is critical to develop suitable inductive biases and learning algorithms that can automatically discover the latent task structure (Boopathy et al., 2025) and filter out the non-compositional features (Jarvis et al., 2024).

Some recent studies took important steps in this direction by identifying useful inductive biases with carefully designed experiments. For example, (Bahdanau et al., 2018) highlighted that a perfect one-to-one mapping between subtasks and modules is not always the best design strategy: architectures composed of modules that specialize in performing groups of related subtasks can perform better, likely because they can learn to identify and leverage commonalities between subtasks. Similarly, (Béna & Goodman, 2021) showed that strong inter-module connection sparsity and resource constraints (measured by the number of units in a module) facilitate module specialization.

### 3.3.2 Continual Learning

While regularization-based approaches (Kirkpatrick et al., 2017; Zenke et al., 2017) and replay-based approaches (Shin et al., 2017; Lopez-Paz & Ranzato, 2017; Rolnick et al., 2019) are popular solutions for tackling catastrophic interference (McCloskey & Cohen, 1989) in continual learning — where learning signals from different tasks interfere with one another — modular architectures are inherently structured to provide a solution to this problem (Parisi et al., 2019; Hadsell et al., 2020; Wang et al., 2024). In fact, architectures with task-specific parameters that are only trained with their corresponding, task-specific learning signals cannot suffer from interference by construction. As a result, several flavors of modularity approaches have been explored and proven valuable in continual learning settings. Many approaches rely on dynamic architectures that learn to perform new tasks based on a partially trainable, shared network — which is trained on previous tasks — and a fully trainable, task-specific module that is dynamically added to the network. Some approaches keep the shared network completely frozen and add fixed-capacity modules (Rusu et al., 2016). Other solutions allow selective fine-tuning of some parameters of the shared network and add modules with a capacity that depends on how much the task differs from the ones that were previously encountered (Yoon et al., 2017); Finally, alternative approaches frame the learning problem as one of selecting, training, and freezing the modules on a task-specific path through a fixed high-capacity network (Fernando et al., 2017). For a recent review of the latest modular and non-modular approaches, we refer the reader to (Wang et al., 2024).

### 3.3.3 Transfer Learning and Large Language Models

Modular approaches are also widely used in transfer learning settings as a way to boost cross-task positive transfer while reducing the number of parameters to fine-tune. This is because it has been found that fine-tuning the entire network on a downstream task is often unnecessary for good performance; instead, training only the last layers or small, task-specific *adapter* modules interspersed within the pre-trained network is typically sufficient (Pan & Yang, 2009; Collobert et al., 2011; Donahue et al., 2014; Zeiler & Fergus, 2014; Rebuffi et al., 2017; Houlsby et al., 2019). This strategy can be adopted, for example, to reuse the feature extractors a CNN learned with extensive training on a large-scale dataset in a new, data-limited domain (Donahue et al., 2014; Zeiler & Fergus, 2014; Rebuffi et al., 2017). More impressively, it can even be used for multi-task cross-lingual transfer, where a pre-trained LLM is repurposed to perform new target tasks in new languages by carefully designing and combining task- and language-specific adapter modules (Pfeiffer

et al., 2020b). For a careful review of modular approaches in transfer learning, we refer the reader to Pfeiffer et al. (2023).

Modularity is also the essential design feature in the emerging LLM frameworks of *Model MoErging* and *Augmented Language Models.* Model MoErging (reviewed in Yadav et al. (2024)) is a new framework designed to build general-purpose AI systems with emergent capabilities by effectively composing independently pre-trained expert models. The compositions are achieved through the careful selection of pre-trained models, as well as routing and aggregation functions. Augmented Language Models — reviewed in Mialon et al. (2023) — are systems composed of LLMs and task-specialized modules, and thus also, clearly, modular. These models can be trained or instructed to leverage external *tools* to solve specific subtasks: for example, they can use a calculator tool to perform arithmetic operations, a retriever tool to retrieve information from document collections, a web browser tool to perform web searches, or a code interpreter tool to execute Python code.

Finally, we note that state-of-the-art LLMs have been increasingly leveraging (Jiang et al., 2024; Guo et al., 2025) or are rumored to leverage[8] Mixture-of-Experts (MoE) layers (Jacobs et al., 1991b) as an efficient way to expand model capacity. One of the first successful applications of MoE in LLMs was the Sparsely-Gated MoE Layer, introduced by Shazeer et al. (2017), which enabled a 1000x increase in model capacity with only a modest rise in computational cost. Building on this, Fedus et al. (2022) proposed the Switch Transformer, which replaces standard feedforward network layers in the Transformer architecture (Vaswani, 2017) with a sparser MoE routing strategy. This approach routes each token to a single expert, achieving a 7x increase in training speed. Finally, recent advances in retrieval techniques (Lample et al., 2019) have led to the development of the Parameter-Efficient Expert Retrieval (PEER) layer, a new MoE variant that offers an improved compute-performance tradeoff and promises to improve training speed further (He, 2024).

### 3.3.4 Autonomous Agents

Similarly, LLM-based agents — autonomous agents that use LLMs as a reasoning engine to flexibly make decisions to interact with their environment — are also, by design, modular. As a matter of fact, LLM-based agents (reviewed in Sumers et al. (2023)) often feature not only an LLM module as a main reasoning engine and external tools, but also additional specialized modules. For example, they are often endowed with additional short- and long-term memory modules — e.g., to keep track of their state and past experiences — learning modules — e.g., to select episodes or insights worth storing — and evaluator modules — e.g., to refine the decisions of the main LLM reasoning module.

Highly modular is also the influential architecture proposed by (LeCun, 2022) for autonomous agents that can learn and exploit world models to reason and plan at multiple levels of abstraction. At the core of the architecture are six interacting, fully differentiable modules: a perception module, a world model module, a short-term memory module, an actor module, a cost module, and a configurator module. The modules can interact in two main working modes, which mirror Kahneman's Dual-Process Theory of Cognition (Kahneman, 2011). During Mode-1, the actor directly computes an action and sends it to the effectors based on inputs from the perception, short-term memory, and configurator modules; this mode is purely reactive and does not involve planning or world model predictions. During Mode-2, the actor infers a minimum-cost action sequence through an iterative optimization procedure involving the world model module — which predicts the likely world state sequence resulting from the proposed action sequence — and the cost module — which computes the costs of the predicted world state sequence. Mode-2 is thus a form of reasoning-based planning that engages the entire system; it is thus computationally expensive and only used when encountering novel tasks. After a Mode-2 engagement episode, the experienced world state sequence and the minimum-cost action sequence can be reused to train the actor to perform amortized action inference. This mechanism ensures that the agent will rely on the cheaper, reactive Mode-1 to deal with future occurrences of the states that led to Mode-2 engagement.

---

[8]https://openai.com/index/gpt-4-research/

### 3.3.5 Hybrid Architectures

Modular designs are also a natural choice for the hybrid architectures typically used for multi-modal AI systems (e.g., Achiam et al. (2023); Anil et al. (2023); Liu et al. (2024a) reviewed in Song et al. (2024); Liang et al. (2024)) as well as for neuro-symbolic architectures (e.g., Mao et al. (2019); Guan et al. (2025), reviewed in (Garcez & Lamb, 2023; Chaudhuri et al., 2021)). Multi-modal models typically learn modality-invariant representations by aligning modality-specific representations of multi-modal data. In these models, the modality-specific representations are often computed with dedicated encoder models (Radford et al., 2021; Alayrac et al., 2022), which can be pre-trained. Similarly, neuro-symbolic architectures — which attempt to combine the strengths of deep learning and symbolic approaches — are also modular by design. These architectures often feature deep learning modules to extract abstract, high-level features from raw, input data and symbolic approaches to perform high-level reasoning using symbolic tools like heuristic search, automated deduction, and program synthesis. This organization — often interpreted as emulating the fast, reactive System-1 thinking and the slow, analytical System-2 thinking (Kahneman, 2011) — is often shown to boost abstract reasoning, interpretability, and safety.

A great example of a multi-modal, neuro-symbolic architecture is the one proposed by Mao et al. (2019), which performs image question answering by leveraging deep learning-based components to learn embeddings of images and questions and symbolic components to perform reasoning on the latent input representations. Specifically, the architecture features three main modules: a perception module, a semantic parser module, and a program executor module. The perception module leverages two specialized CNN architectures (He et al., 2017; 2016) to extract latent representations of the visual attributes of all the objects in the scene. The semantic parser module translates the input question into an executable program in a target domain specific language (DSL) using two feed-forward and two recurrent GRU networks (Cho et al., 2014). Finally, the program executor module generates the final answer by executing the executable program on the image embeddings.

## 4 Modularity in Brains

In the previous sections, we illustrated why brains, like most complex systems, are modular (§ 2.2), and what advantages this property provides (§ 2.1). Here, we discuss different, selected perspectives on brain modularity. In recent years, one of the most influential modular decompositions of the brain's cognitive abilities in AI has been Kahneman's Dual-Process Theory of Cognition (Kahneman, 2011; LeCun, 2022; Goyal & Bengio, 2022), which posits the existence of two complementary systems underlying human cognition: a fast, reactive system underlying intuition and a slow, analytical system underlying reasoning. However, this is only part of the picture. Brains are hierarchically modular (Meunier et al., 2009), that is, modular at different spatial scales and levels of abstraction. While there is no consensus on which analysis level best captures the fundamental principles underlying human intelligence, each perspective provides valuable insights. Here, we review several influential perspectives spanning these different levels.

### 4.1 Neurons as Fundamental Modules

At the lowest spatial scale, we have *neurons*. While neurons in standard feedforward networks compute a simple weighted sum of their inputs followed by a nonlinearity (McCulloch & Pitts, 1943), biological neurons exhibit far greater complexity (Beniaguev et al., 2021). Each biological neuron operates as a multi-state, nonlinear dynamical system (Hodgkin & Huxley, 1952) that generates binary signals, or spikes, whose precise timing carries behaviorally relevant information (Maass, 1997). Moreover, rather than merely summing inputs linearly, biological neurons process incoming signals nonlinearly along their dendritic branches. In fact, a single neuron typically receives multiple synaptic connections from each presynaptic source, with dendritic integration introducing additional layers of potentially different nonlinear processing before signals reach the soma (London & Häusser, 2005; Jones & Kording, 2020). This suggests that a single biological neuron can be thought of as a recurrent, highly nonlinear, multilayer network in itself — one endowed with inductive biases that enable it to extract rich, structured features from its inputs. Interestingly, recent work (Liu et al., 2024b) has demonstrated promising results on AI tasks by introducing greater flexibility in the nonlinear functions that neurons use to process their inputs.

### 4.2 Canonical Microcircuits

Moving up the hierarchy, we encounter *canonical microcircuits* (Harris & Shepherd, 2015). Anatomical studies have shown that the cortex is systematically organized into six layers (or laminae), labeled L1 through L6, running parallel to the skull. Electrophysiological analyses have further revealed the existence of stereotyped synaptic connectivity patterns, which induce recurrent loops between neurons in these layers. These network motifs are ubiquitous across the cortex, encompassing motor, visual, somatosensory, and auditory areas (Douglas & Martin, 2004). One of the most well-established loops begins in the thalamus, which provides input to L4. From there, information is relayed to L2/3, which in turn excites L5/6 of the same cortical area. L5/6 neurons follow two main pathways: one loops back to L4, forming an inner recurrent circuit, while the other projects to subcortical regions, including the thalamus, forming an outer feedback loop. Additionally, an inter-area loop has been identified, induced by L2/3 neurons projecting to L4 of adjacent cortical areas, facilitating cross-regional communication. Canonical microcircuits are thought to play a crucial role in integrating sensory inputs relayed through the thalamus with contextual information from other cortical areas, enabling context-dependent decision-making (Haeusler & Maass, 2006). These circuits have been linked to predictive coding theories (Bastos et al., 2012), which propose that the brain computes prediction errors to refine future inferences and minimize surprise. They are also the fundamental module in the Thousand Brains Theory of Intelligence (Hawkins, 2021), which suggests that cortical columns — comprising preferentially connected neurons spanning all six cortical layers within a cylindrical region — learn sensory-input-dependent models for the objects we interact with. However, it is important to note that many other network motifs and recurrent loops exist throughout the brain, some of which are relatively frequent but remain less well understood (Shepherd & Yamawaki, 2021). Do these stereotypical connectivity patterns confer any computational advantages? Recent influential work (Chen et al., 2022) suggests that they may enhance out-of-distribution generalization. By initializing a recurrent network's weights based on connectivity patterns observed in the primary visual cortex (Billeh et al., 2020), the study demonstrated significantly improved OOD generalization compared to both feedforward and recurrent CNNs.

### 4.3 Cortical Areas and Cortical Networks

Finally, at the highest spatial scale, we find *cortical areas* and *cortical networks*. Cortical areas are identified by parcellating the cortex into clusters of adjacent neurons with shared common properties such as histological characteristics, connectivity patterns, spatial tuning, and functional tuning (Van Essen & Glasser, 2018; Petersen et al., 2024). These parcellations are typically obtained from data collected using invasive methods and processed with varying assumptions and algorithms. Consequently, estimates of the number of cortical areas vary widely, generally ranging from 100 to 200 regions, with no universal consensus. However, a recent semi-automated, multimodal parcellation has gained traction, identifying 180 areas per hemisphere (Glasser et al., 2016). An alternative emergent modular decomposition approach leverages non-invasive functional magnetic resonance imaging (fMRI) recordings. These data are used to identify *functional networks*, which consist of multiple cortical regions that tend to be coactivated during the execution of cognitive tasks or while at rest. Unlike anatomy-based parcellations, these networks can include spatially distant regions. Although there is no universally accepted functional parcellation, influential studies have decomposed the cortex into 7 to 20 large-scale networks (Yeo et al., 2011; Power et al., 2011). Importantly, a recent meta-analysis (Uddin et al., 2019) examined the commonalities among functional networks identified in multiple resting-state and task-based fMRI studies and identified six core functional networks, each linked to distinct cognitive functions. These comprise: (1) the *occipital* network, involved in visual processing; (2) the *pericentral* network, supporting somatomotor functions; (3) the *dorsal frontoparietal* network, mediating attentional control; (4) the *lateral frontoparietal* network, regulating executive control; (5) the *midcingulo-insular* network, controlling salience; (6) the *medial frontoparietal* network, responsible for the default mode of brain activity.

### 4.4 Behavioral Decomposition of Cognitive Abilities

So far, we have discussed modular decompositions of human intelligence based on direct neural recordings. An alternative approach relies on the analysis of large-scale cognitive test results (Kaufman, 2018). The most

influential framework emerging from this approach is the Cattell-Horn-Carroll (CHC) theory (Schneider & McGrew, 2012), which models intelligence as a three-tier hierarchical structure based on the analysis of correlation patterns among test scores. At the lowest level are *narrow* abilities—specialized cognitive skills applied across multiple tasks. At the intermediate level are *broad* abilities—general cognitive functions that encompass multiple narrow abilities. For instance, Kaufman (2018) identified 17 broad abilities. At the highest level is the *g-factor* (Jensen, 1998), which contributes to all broad abilities and has been associated with brain properties such as size, energy efficiency, nerve conduction velocity, and inter-node path length (Deary et al., 2010). Computational problem-solving, for example, is believed to depend on three broad abilities: fluid reasoning, perceptual processing, and short-term memory (Román-González et al., 2017; Ambrósio et al., 2014). Fluid reasoning, in turn, comprises three narrow skills: *inductive reasoning —* the ability to infer underlying patterns or rules from observed data; *general sequential reasoning —* the ability to apply learned rules sequentially to solve problems; and *quantitative reasoning —* the ability to use mathematical relationships and operations to reason about numerical quantities.

### 4.5 Phylogenetic Identification of Brain Modules

Finally, we note that all the approaches discussed thus far attempt to identify brain modules based on direct or indirect measurements of the brain's current state. However, an alternative perspective argues that a true modular decomposition of the brain requires integrating phylogenetic data to study its evolutionary history (Cisek, 2019; Cisek & Hayden, 2022). This approach aims to infer the sequence of modifications that transformed primitive feedback control mechanisms—originally implemented by single cells in multi-cellular organisms to maintain homeostasis—into the complex decision-making systems of the mammalian brain. The central idea is that neural modules emerged and evolved incrementally, adapting to an ever-changing environment by enabling progressively more sophisticated behaviors. Consequently, understanding the present function of a neural module requires examining the role it played at the time of its evolutionary emergence. This approach has already led to consolidated theories about the evolution of selected brain modules (Feenders et al., 2008; Chakraborty & Jarvis, 2015; Kebschull et al., 2020) and to increasingly detailed descriptions of the evolutionary origin of the existing modular organization (Cisek, 2019; Cisek & Hayden, 2022)

## 5 Open Questions

### 5.1 Modularity or Scaling? Competing Paths for AI Progress

In the previous sections, we have reviewed a substantial body of research that highlights the critical role of modularity in designing intelligent systems. However, it may also be argued that modular architectures and architectural constraints could, in fact, hinder AI system performance and that research should instead prioritize scaling model and data size. While this view seemingly contradicts classical results like the No Free Lunch Theorem (Wolpert et al., 1995), it aligns with the *Bitter Lesson* (Sutton, 2019), which suggests that progress is driven by scaling rather than handcrafted design. Scaling laws (Kaplan et al., 2020) further support this perspective, showing that performance consistently improves with larger models and datasets. While this finding is supported by a large body of work, a scale-centric approach has significant downsides. Expanding model and dataset size amplifies financial costs and environmental impact, both of which are already pressing concerns (Bender et al., 2021). Moreover, state-of-the-art AI systems have nearly exhausted publicly available internet data, raising concerns about data limitations. While synthetic data generation shows promise (Singh et al., 2024), it risks leading to model collapse (Shumailov et al., 2024). Given these constraints, prioritizing data quality over sheer quantity is becoming increasingly important. In fact, high-quality data can reduce dataset size and training time while matching (Eldan & Li, 2023) or even surpassing larger models trained on less curated data (Gunasekar et al., 2023)

### 5.2 Are Brains Still Relevant for AI Development?

Given the impressive capabilities of current AI systems, does it still make sense to look to the brain as a source of inspiration for improving AI? Although it is sometimes argued that AI systems have already sur-

passed human capabilities, such achievements have so far been limited to narrow, well-defined tasks. General intelligence—characterized by the ability to reason, plan, and act flexibly across diverse domains—remains out of reach. For instance, while large language models (LLMs) may rival—or even surpass—human performance in *formal* language skills, they continue to fall short in *functional* linguistic abilities (Mahowald et al., 2024). Additionally, human intelligence encompasses far more than language processing: humans can seamlessly integrate perception, reasoning, and action within a single, adaptable system. Moreover, as noted in § 1, current AI models also require orders of magnitude more data and computation than the human brain to achieve comparable performance. Taken together, these observations underscore that human intelligence remains a meaningful benchmark for AI, and that biological brains continue to provide valuable insights for AI research, as they have historically. The development of AI and deep learning has, in fact, long been intertwined with efforts to understand and emulate human cognition (Appendix A), and several brain-inspired principles have already contributed to significant advances in the field (e.g., LeCun et al. (1998); Kirkpatrick et al. (2017); Rolnick et al. (2019)).

### 5.3 How Can Modularity Help Bridge the Gap Between Natural and Artificial Intelligence?

Identifying and translating principles from neuroscience into useful inductive biases for AI systems remains a significant challenge. Should we attempt to replicate the brain's connectivity patterns, its activation dynamics, or even its reliance on spike-based communication? We suggest that modularity offers a promising framework for addressing this challenge. One potential path forward is to draw on neuroscientific findings to identify the functional modules that the brain employs to solve specific tasks and implement analogous modules in AI systems. For instance, it is well established that memory and language are supported by distinct systems in the brain (Squire, 2009). While conventional transformer-based LLMs do not include a dedicated memory module, even simple implementations of memory-augmented LLMs have demonstrated benefits, such as reduced hallucination rates and parameter counts (Shuster et al., 2021; Borgeaud et al., 2022; Lewis et al., 2020)—key factors behind the popularity of the retrieval-augmented generation (RAG) framework in industrial chatbots—and improved compositional generalization abilities (Lake, 2019). Similarly, emerging evidence suggests that language and thought are supported by distinct neural systems in the brain (Fedorenko et al., 2024; Mahowald et al., 2024). In contrast, current LLMs do not employ a dedicated reasoning module; rather, reasoning behavior appears to emerge implicitly from training procedures that encourage the model to generate more output tokens before producing a final answer (e.g., Guo et al. (2025)). This raises an important question: could an architecture that explicitly separates language and thought processes lead to improved reasoning capabilities? How might a dedicated reasoning module be designed, and which features of the brain's reasoning mechanisms would be most beneficial to emulate? These remain open and challenging questions. Moving toward a deeper alignment with the brain's structure—particularly at Marr's algorithmic and implementational levels (Marr, 2010)—is considerably more complex, and it remains uncertain whether such efforts would yield clear practical advantages. Nonetheless, as illustrated above, we believe that the computational level offers a viable framework for identifying and transferring functional modules—and, through them, core features of human cognition. In summary, given the demonstrated benefits of incorporating modularity principles into AI architectures (§ 3) and the consistent evidence that such principles are exploited by the brain (§ 4), modularity appears to offer an effective framework for integrating brain organization principles into AI system design. This approach holds considerable promise for enhancing the efficiency and performance of artificial systems; however, further research is needed to determine how to best implement these principles in practice.

## 6 Conclusion

Modular designs are ubiquitous across both natural and artificial systems and have been explored—often in isolation—within distinct subfields of biology and engineering. In this work, we brought together these diverse threads through a unified framework and argued that modularity is not merely a convenient structural choice or a biological epiphenomenon but a fundamental principle underlying both natural and artificial intelligence.

We examined the computational advantages modularity offers through multiple perspectives, ranging from engineering and evolutionary biology to artificial intelligence and systems neuroscience. We reviewed how

modular structures are implemented—explicitly or implicitly—in contemporary AI architectures and how modularity principles characterize the organization of the brain at several levels of abstraction, from microcircuits to large-scale cognitive systems. Moreover, we discussed how modularity may serve as a viable framework for integrating brain-inspired principles into AI system design. For instance, one promising approach is to identify the functional modules that the brain leverages to solve specific tasks—such as the dedicated systems supporting memory and reasoning—and implement analogous components in AI architectures. This integration can mitigate some of the critical challenges that current AI systems face, including high hallucination rates, poor sample efficiency, limited generalization, and high energy demands. The brain, shaped by hundreds of millions of years of evolution, offers an exceptionally efficient and adaptable model—one that continues to provide insights for improving artificial systems. Despite the promise, significant challenges remain. These include identifying the appropriate level of abstraction to extract functionally relevant brain modules, determining how to translate those modules into trainable or hardwired components in neural architectures, and designing mechanisms for effectively integrating their outputs.

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

# A    Approaches to Brain-Inspired Artificial Intelligence

## A.1    Historical Foundations

Artificial Neural Networks (ANNs) can be traced back to efforts by McCulloch and Pitts aimed at under-standing how networks of biological neurons could compute logical functions (McCulloch & Pitts, 1943). Similarly, the birth of Artificial Intelligence as a research field — typically traced back to the famous 1956 Dartmouth Workshop organized by McCarthy, Minsky, Rochester, and Shannon (McCarthy et al., 2006) — was driven by initiatives intended to prove that a thorough, computational description of human learning and intelligence could lead to the creation of machines that can simulate such natural processes. Thus, clearly, a desire to understand and model the brain and its unique capabilities was a major goal underlying the creation of the first ANNs and the AI research field. The idea of using the brain as a source of inspiration for designing more robust, reliable, and performant Deep Neural Networks (DNNs) has also been influential (though not central) in AI in the more recent past. Over the last decade, a few review papers have discussed this perspective from different angles. These papers identified the most evident limitations of DNNs that emerge from comparisons with humans and elaborated on different ways the brain can be harnessed as a guide to bridge the gap. Hassabis et al. (2017) highlighted how the history of deep learning is strongly inter-twined with that of neuroscience and how different neural features have profoundly impacted deep learning research or are poised to do so. For example, visual cortical neurons' tuning and normalization properties directly inspired CNNs. Similarly, efforts to model animal conditioning and experience replay profoundly impacted RL, while synaptic plasticity phenomena inspired several influential continual learning algorithms. Conversely, other active research areas, such as efficient learning, transfer learning, and long-term planning, stand to gain from yet unexplored neural features.

## A.2    Influential Perspectives on Brain-Inspired AI

As discussed in the main sections of this paper, modularity is a fundamental feature of both natural and artificial intelligence and offers one promising route for bridging the two. Other authors, however, have proposed influential alternative approaches. For example, Zador (2019) argued that to understand the brain and improve deep learning models, we should focus on identifying the fundamental *wiring rules* that are encoded in the 1 GB human genome; such rules are critical as they must provide a highly compressed representation of the entire 200 TB[9] human connectome (Seung, 2012), distilling knowledge acquired over evolutionary timescales into brain networks that support critical innate behaviors and fast learning. Taking a more pragmatic approach, Richards et al. (2019) recommended identifying the three main computational building blocks of the brain: its *cost functions* — which reflect the networks' learning goals — its *optimization algorithms* — which guide synaptic plasticity — and its *backbone architectures* — which constrain how the information can flow across the network. Sinz et al. (2019) went one step further and suggested a purely data-driven approach to boosting the generalization performance of DNNs based on aligning the latent features DNNs learn with the recorded brain activity patterns.

While the prevailing view in AI is that we should aim to directly try to reproduce the high-level cognitive abilities unique to human adults to advance the development of AI systems, more recent work (Zaadnoordijk et al., 2022; Zador et al., 2023) has emphasized a different perspective. For instance, Zaadnoordijk et al. (2022) highlighted the importance of examining infants' learning. Studies have shown that the infant brain already possesses adult-like structural and functional connectivity patterns, allowing it to perform efficient multimodal, unsupervised learning by exploiting attentional, processing, and cognitive biases as well as curriculum and active learning strategies. Interestingly, this view is consistent with previous observations (Lake et al., 2017) stressing the importance of infants' *start-up software* consisting of causal world models and intuitive theories of psychology and physics that boost compositional meta-learning. In a similar vein, Zador et al. (2023) advocated a focus on reproducing animal-level intelligence first, as animals already possess a vast amount of developmentally inherited knowledge that allows them to thrive in their constantly changing environment through state-dependent decision-making and detailed world models.

---

[9]Assuming $10^{14}$ synaptic weights stored in half-precision floating-point format (FP16)

### A.3 Modularity-Centered Perspectives

The revised work above offers interesting perspectives on how selected brain properties can guide the development of new models that move beyond narrow AI systems. However, none of them attempted to decompose brains into their fundamental functional building blocks, or modules, underlying intelligence. A notable exception is provided by recent work (Marblestone et al., 2016; Goyal & Bengio, 2022; Mahowald et al., 2024) that represents a significant step in this direction. Marblestone et al. (2016) took an optimization-centric approach and tried to organize the brain in terms of cost functions it appears to optimize. Specifically, they surveyed different AI-relevant functions the brain is known to perform effectively — including high-level planning, hierarchical predictive control, short- and long-term memory, selective attention, and information routing — attempted to map these functions to specific brain networks, and hypothesized how these networks might be coordinated to learn over different timescales. Goyal & Bengio (2022) attempted to identify the core cognitive principles of human intelligence and to suggest potential ways to translate them into inductive biases for deep learning architectures, building on influential cognitive neuroscience theories, such as the Global Workspace Theory (GWT) (Baars, 1997) and the Dual-Process Theory of Cognition (Kahneman, 2011). Mahowald et al. (2024) focused on LLMs and explained their inconsistent performance across different tasks as arising from a clear-cut separation of strictly linguistic, formal *abilities* — which they excel in — from higher-level, *functional abilities* — which they often struggle with. Neuroscientific evidence indicates that these abilities are supported by distinct brain networks, suggesting that architectural and emergent modularity approaches, which mirror the specialization observed in the brain, are essential for enhancing the capabilities of LLMs.

