# OpenReview forum: "Modularity is the Bedrock of Natural and Artificial Intelligence"
_TMLR — Rejected by TMLR_

### Review · Reviewer_pZnt · 2025-06-19

**Summary Of Contributions:**

This paper aims to make the case that modularity is a fundamental principle underlying the structure of complex systems including human brain and human intelligence, and therefore, this principle should also be adopted for the construction of artificial intelligence. The paper makes its argument by citing by many papers from diverse areas of study.

**Audience:**

Yes

**Claims And Evidence:**

No

**Requested Changes:**

The paper might be more effective in conveying its ideas if it were presented as a survey paper. The paper could be structured around two central questions: (1) How has the principle of modularity been applied in modern artificial intelligence?; (2) How can the different approaches to building modular AI be best organized so as to guide future research?

I believe the current Section 3 of the paper is a good initial attempt at answering these questions. Significantly expanding this section at the cost of other sections would be helpful.

**Strengths And Weaknesses:**

### Strengths
1. The paper cites many interesting references from diverse areas of study. It can be a good resource for someone interested in reading about modularity from a systems perspective.
2. The paper's characterization of modularity in aritifical intelligence in Section 3 can be a useful starting point for this discussion.

### Weaknesses
1. The paper does not present concrete new ideas on how to construct artificial intelligence systems in a modular fashion.

2. The paper is written like a position paper but it does not make new arguments in favor of its position.

---

> ### Author Response · Authors · 2025-09-07
>
> We thank the reviewer for their thoughtful comments and appreciate their constructive feedback in particular. We are pleased that they recognized the paper’s value in bringing together a diverse range of references across various fields, making it a valuable resource for readers interested in modularity from a systems perspective. We also appreciate their observation that our discussion of modularity in artificial intelligence provides a helpful entry point into this topic.
>
> Regarding the concern that the paper resembles a position piece without presenting new arguments in support of its stance, we would like to clarify that our article is intended as a survey that draws new connections, rather than a position paper. As outlined in TMLR’s Submission Guidelines and Editorial Policies (https://jmlr.org/tmlr/editorial-policies.html), surveys are expected to “draw new connections, highlight trends, and suggest new problems in an area.” Our contribution lies in synthesizing work from diverse fields to identify modularity as a unifying principle underlying intelligence in evolutionary biology, neuroscience, and artificial intelligence. While modularity has been studied extensively within each of these domains in isolation, our paper takes it a step further by demonstrating that it represents a fundamental computational principle that transcends them, thereby offering a new integrative perspective.
>
> Regarding the point that the paper does not introduce new methods for building modular AI systems, we agree that developing such approaches is a significant challenge; however, we emphasize that it lies beyond the scope of our work. In Section 5, we did provide some indirect evidence that brain modularity principles could help bridge the gap between natural and artificial intelligence (for instance, the recent success of RAG in LLMs can be viewed as a step toward modular memory). We also outlined some possible future directions that could be followed to further narrow this gap, based on identifying and implementing additional functional modules of natural intelligence (some of which are discussed in Section 4) or learning systems (discussed in Appendix A). That said, we acknowledge that developing concrete methods requires extensive theoretical and empirical work, which is beyond the scope of the present paper. To clarify these points, we have revised Section 5 to make it more explicit that this is an open question for future work.

---

### Review · Reviewer_AHZR · 2025-06-23

**Summary Of Contributions:**

This paper surveys modularity both in artifical intellignece and neurosicence, by exploring how it has emerged as a solution in several AI research areas and how it can help bridge the gap between natural and artificial intelligence.

**Audience:**

No

**Broader Impact Concerns:**

No ethical issues.

**Claims And Evidence:**

Yes

**Requested Changes:**

see weaknesses

**Strengths And Weaknesses:**

Strengths:
1. The modularity in artifical intellignece and neurosicence is a very interesting topic.

Weaknesses:
1. Authors can propose more novel claims to complete this survey, exsiting conclusions are too common.

2. Some recent works are missing, such as Modular for LLMs, brain-inspired AI, nurosymbolic AI, etc.

3. In section 3.3, modularity for AI can be introduced with more technical depth, and more novel insights.

---

> ### Author Response · Authors · 2025-09-07
>
> We appreciate the reviewer’s raised points and are pleased that the relevance of the paper’s topic was recognized
> .
> Regarding the novelty of our claims, we would like to clarify that the paper is intended as a survey article that draws new connections, in line with TMLR’s Submission Guidelines and Editorial Policies (https://jmlr.org/tmlr/editorial-policies.html) on survey papers. Our goal was not to introduce entirely new methods, but rather to synthesize work across evolutionary biology, neuroscience, artificial intelligence, and engineering theory in order to highlight modularity as a unifying computational principle underlying intelligent systems. We have addressed this point more fully in our general revision comment and have revised the paper to clarify its structure, including a more explicit division into sections and subsections, a table of contents, and a new section in the introduction that clarifies the paper's aims.
>
> Regarding the concern about insufficient discussion of topics such as modularity in LLMs, brain-inspired AI, and neurosymbolic AI, we note that these areas are addressed in the paper. Specifically, Section 3.3.3 discusses modularity in LLMs, Section 3.3.4 addresses modularity in LLM-based agents, and Section 3.3.5 covers neurosymbolic architectures. Section 4 further considers high-level strategies for bridging the gap between natural and artificial intelligence, emphasizing that this remains an open research problem. Finally, Appendix A provides additional detail on modular and non-modular approaches to brain-inspired AI.

---

### Review · Reviewer_Mzox · 2025-08-24

**Summary Of Contributions:**

This paper gives a review of modular structures in artificial intelligence systems and biological systems, highlighting the importance of modularity in compositional generalization, continual learning, and transfer learning. The paper also gives a formal definition of modularity, and discusses different types of modularity such as implicit modularity and emergent modularity

**Audience:**

No

**Claims And Evidence:**

Yes

**Requested Changes:**

Please refer to the Weaknesses above. I think in the first place, the paper needs some clarification on its purpose and takeaway messages.

**Strengths And Weaknesses:**

Strengths:

1. The paper focuses on an important topic: the modularity in engineering and nature.

2. The paper tries to unify the modularity in AI systems and human brains.


Weaknesses:

1. I'm not sure whether this paper is a survey or a position paper. It is not clear from the current manuscript. It would be better if the authors can clarify the purpose of writing this paper.
- It looks like a survey because it does a (relatively) comprehensive review of modularity-related papers. However, the paper lacks a clear taxonomy and roadmap that are generally required for a survey.
- It also looks like a position paper because of its title and strong claims to advocate for paying more attention to modular designs. However, it does not provide much new insight beyond literature reviews.

2. The main takeaways from this paper are quite unclear. The paper seems to cover a wide range of literature that discusses modularity in different contexts, but does not give in-depth analysis of them. I have expected to see messages or insights such as "how to design neural networks that can have higher modularity with little performance drop" or "how to better encourage the emergence of modularity during training," but there are no such takeways,

3. The paper has no figures or tables. It is highly recommended to have figures or tables that summarize the high-level story of the whole paper.

---

> ### Author Response · Authors · 2025-09-07
>
> We thank the reviewer for their constructive feedback and particularly appreciate that they recognized modularity as an important topic in both engineering and nature, and that they saw our work as a useful attempt to highlight its role across brains, machines, and learning paradigms.
>
> First, we acknowledge that the initial version of the paper may not have made it clear whether it should be read as a review or a position paper. To clarify, our article is intended as a survey that draws new connections, consistent with TMLR’s Submission Guidelines and Editorial Policies (https://jmlr.org/tmlr/editorial-policies.html), which describe surveys as contributions that “draw new connections, highlight trends, and suggest new problems in an area.” We have addressed this point more fully in our general revision comment and have revised the paper to clarify its structure, including a more explicit division into sections and subsections, a table of contents, and a new section in the introduction that clarifies the paper's aims.
>
> Second, we agree that developing new methods to increase modularity and improve performance is an important research direction, and we personally believe that brain modularity principles may play a crucial role in this endeavor. Although we suggested in Section 5 how bridging this gap could be accomplished at a high level, we believe that rigorously defining and testing new methods requires dedicated theoretical and experimental work, which lies beyond the scope of this paper. Instead, our goal was to review literature across multiple scientific domains and identify a common thread: that modular organization is not just a recurring theme in different scientific fields but likely a fundamental computational principle of complex systems that both brains and machines leverage.
>
> Third, regarding the observation that the manuscript was too text-heavy, we have more clearly divided the paper into sections and subsections and added a table of contents. We are also working on tables and figures to highlight and summarize the main topics, which we believe will further improve readability and accessibility.

---

### Decision · Action_Editor_J3Qy · 2025-10-06

**Recommendation:** Reject

**Additional Comments:**

Overall, I can see the paper being resubmitted and accepted. But I think the author should either be more bold, and follow through their reasoning on the implications of these different framing, or make the paper into a survey.

**Audience:**

No

**Audience Explanation:**

This is a bit harder to judge. But after reading the reviews and looking at the paper myself I find myself siding with the reviewers. The claims brought up in the paper are not questionable. The bigger issue is the utility of putting together these different threads. I believe the paper could work if either it tries to follow through, and explain how these different roles that modularity has in different subfields might lead to something new. I.e. extrapolate, or show how a particular role it plays (say in neuroscience) it can be a driving force in a different field (e.g. in ML). But overall I find that the paper does not make that necessary step to become useful. To inspire new research directions, or propose some roadmap that others can follow.

Overall I do think that a more likely utility of the work is as a survey providing all the relevant citations in one place. However the paper is not really framed like that. Making it confusing. I think the author can maintain the current framing, but they need to be more bold and bring this different perspectives on modularity together beyond just having them listed in different chapters of the paper.

**Claims And Evidence:**

Yes

**Claims Explanation:**

The paper is somewhere between a position paper and a survey. The claims made by the paper is sustained by citing relevant literature. The claims are typical in the literature where they are introduced, therefore the citations provided are more than sufficient to support the claim. The value of the paper is meant to be the fact that it brings together observations made in various subfields.

**Resubmission Of Major Revision:**

The authors may consider submitting a major revision at a later time.